# COVID-19-Associated Myocarditis: An Evolving Concern in Cardiology and Beyond

**DOI:** 10.3390/biology11040520

**Published:** 2022-03-28

**Authors:** Meg Fraser, Arianne Clare C. Agdamag, Valmiki R. Maharaj, Melinda Mutschler, Victoria Charpentier, Mohammed Chowdhury, Tamas Alexy

**Affiliations:** 1Department of Medicine, Division of Cardiology, University of Minnesota, Minneapolis, MN 55455, USA; mfraser10@umphysicians.umn.edu (M.F.); agdam001@umn.edu (A.C.C.A.); maha0104@umn.edu (V.R.M.); mmutschl10@umphysicians.umn.edu (M.M.); 2Department of Medicine, University of Minnesota, Minneapolis, MN 55455, USA; charp024@umn.edu; 3Cardiology, North Central Heart, Sioux Falls, SD 57108, USA; mchowdhury@ncheart.com

**Keywords:** COVID-19, myocarditis, biomarkers, cardiac imaging

## Abstract

**Simple Summary:**

Coronavirus disease-2019 (COVID-19) affects many organs in the body, including the heart. One complication of particular concern is inflammation of the heart muscle, called myocarditis. This paper presents updated research data on COVID-19-associated myocarditis. Specifically, we review the incidence, potential mechanisms, blood and imaging tests that can be used to detect the disease. We emphasize that, in contrast with early reports, recent data suggest that myocarditis in the setting of COVID-19 is relatively uncommon, yet infected individuals are at a substantially increased risk for poor outcomes. It is important to continue research in this area.

**Abstract:**

The direct and indirect adverse effects of SARS-CoV-2 infection on the cardiovascular system, including myocarditis, are of paramount importance. These not only affect the disease course but also determine clinical outcomes and recovery. In this review, the authors aimed at providing an update on the incidence of Coronavirus disease-2019 (COVID-19)-associated myocarditis. Our knowledge and experience relevant to this area continues to evolve rapidly since the beginning of the pandemic. It is crucial for the scientific and medical community to stay abreast of current information. Contrasting early reports, recent data suggest that the overall incidence of SARS-CoV-2-associated myocarditis is relatively low, yet infected individuals are at a substantially increased risk. Therefore, understanding the pathophysiology and diagnostic evaluation, including the use of serum biomarkers and imaging modalities, remain important. This review aims to summarize the most recent data in these areas as they relate to COVID-19-associated myocarditis. Given its increasing relevance, a brief update is included on the proposed mechanisms of myocarditis in COVID-19 vaccine recipients.

## 1. Introduction

Coronavirus disease-2019 (COVID-19) is caused by the highly contagious Severe Acute Respiratory Syndrome Coronavirus-2 (SARS-CoV-2). This novel enveloped RNA beta-coronavirus was initially isolated in Wuhan City, China in December 2019 and was identified to cause severe pneumonia in a cluster of patients [1]. It spread around the globe within a short timeframe, and the World Health Organization (WHO) declared COVID-19 a pandemic in March 2020 [2]. Evolving variants of the virus have infected and re-infected the population in a series of “surge waves” with over 380 million laboratory-confirmed cases recorded to date around the world [3]. The estimated worldwide COVID-19-related death toll has surpassed an astonishing 5.6 million as of February 2022 and continues to increase [3].

The manifestations of SARS-CoV-2 infection may present on an extremely broad clinical spectrum. In a substantial proportion of cases, individuals may remain completely asymptomatic, potentially even unaware that they are affected [4]. While the majority of symptomatic patients present with mild complaints, others are hospitalized with hypoxia, respiratory distress, or multi-organ failure associated with extreme mortality risk despite aggressive medical interventions [5,6,7]. Approximately one-third to one-fourth of hospitalized individuals require intensive care unit (ICU) admission, primarily for the management of refractory hypoxic respiratory failure in the setting of acute respiratory distress syndrome (ARDS) [8,9,10]. While the key target of SARS-CoV-2 is the respiratory system, extrapulmonary disease manifestations and complications are also frequent. The most common symptoms include persistent fever, chills, dry cough, dyspnea, congestion, sore throat, fatigue, nausea, emesis, diarrhea, anosmia, ageusia, or any combination of these [5,11,12,13,14]. Involvement of the cardiovascular, hematologic, neurologic, and endocrine systems are also frequent, further obscuring patient presentation and complicating management. In addition, the primary infection may be accompanied by a poorly defined immune-mediated systemic hyperinflammatory response by the host, often referred to as cytokine storm. This is characterized by fever, tachycardia, tachypnea, and refractory hypotension in the setting of severely elevated serum tumor necrosis factor (TNF) and interleukin-6 (IL-6) levels and portends a poor clinical outcome [15,16,17,18].

The direct and indirect adverse effects of SARS-CoV-2 infection on the cardiovascular system are of paramount importance and may significantly hinder outcomes and patient recovery. One of the reported potential mechanisms of myocardial injury in this setting is acute myocarditis. Given the significant research efforts and resources dedicated to this new field, our scientific knowledge is evolving at a rapid pace. Therefore, this review aims to provide an update on the incidence, proposed pathophysiology, and diagnostic evaluation of COVID-19-associated myocarditis.

## 2. Incidence and Clinical Relevance of Myocarditis in the Setting of COVID-19 Infection

Several case reports and case series have been published describing COVID-19-associated myocardial injury and myocarditis with multiple review articles summarizing the key findings [19,20,21]. These studies established a male predominance with a median age of around 50 years. Cardiac biomarkers were shown to be elevated in almost all affected individuals with cardiac involvement portending a significantly increased risk of death. However, it remained challenging to establish the exact prevalence of myocarditis among patients with COVID-19 disease as these early descriptive reports were limited in case numbers. In addition, appropriate diagnostic modalities were infrequently used as hospital services were overwhelmed and procedures were limited to emergency cases to minimize staff exposure [22]. Murk and colleagues subsequently analyzed a United States (US) health claims database using ICD-10 codes to calculate risk estimates and odds ratios of disease association with SARS-CoV-2 infection [23]. Authors found that myocarditis was strongly associated with COVID-19 but overall posed a low absolute risk (OR: 8.17; absolute risk: 0.1%). A follow-up study from Israel confirmed that the risk of myocarditis was substantially increased in patients with COVID-19 (risk ratio: 18.75) [24]. Subsequently, the Centers for Disease Control and Prevention (CDC) published the results of a similar analysis utilizing a large, administrative database including over 900 hospitals within the US [25]. They found that inpatient encounters for myocarditis increased by 42.3% in 2020 compared to 2019 and calculated an absolute risk of 0.146% among patients diagnosed with SARS-CoV-2 infection during an inpatient or hospital-based outpatient encounter. Males and those above 50 years of age were more commonly affected, as were children under age 16 years [25]. After adjusting for patient and hospital characteristics, those with COVID-19 had a 15.7-fold increased risk for myocarditis compared to patients without the disease. A more recent retrospective analysis of over 700,000 patients with SARS-CoV-2 infection found that 5% developed new-onset acute myocarditis that was associated with a significantly increased all-cause mortality at six months [26]. Notably, however, reported clinical outcomes have improved two years into the pandemic, potentially as a result of the aggressive cardiovascular management strategies implemented and the novel medical therapies developed [27,28]. One recently published meta-analysis evaluated the rate of myocarditis in COVID-19 fatalities and suggested that the incidence of myocarditis was less than 2% [29]. While the data and our scientific knowledge continue to evolve rapidly, these reports have established that patients with SARS-CoV-2 infection are at a substantially increased risk to develop myocarditis. They also underscore the need for further investigations to better understand disease pathophysiology and to develop improved diagnostic, management, and preventive strategies.

## 3. Pathophysiology of COVID-19-Associated Myocarditis

The exact pathophysiology of COVID-19-associated myocarditis has not been completely elucidated yet, but multiple potential mechanisms have been proposed. SARS-CoV-2 may affect the myocardium via direct injury or indirectly as a result of innate immune system overactivation [30,31,32]. Initially, the virus enters the cytoplasm of host cells, a process facilitated by binding of the spike (S) glycoprotein, abundant on the surface of SARS-CoV-2, to the angiotensin-converting enzyme 2 (ACE2) transmembrane proteins [33,34,35]. ACE2 is expressed extensively on the cilia of respiratory epithelium, making it a prime target for viral invasion over an extensive surface area [36]. The process is facilitated by transmembrane serine protease 2 (TMPRSS2) and neuropilin 1 (NRP1) [37]. After gaining access to the host cell’s cytoplasm, the viral RNA replicates, translates into proteins, and virion particles are assembled. Subsequently, these are released into the blood and use the circulatory system to propagate the infection to remote organs [38]. ACE2 is known to be expressed on the surface of cardiac myocytes and endothelial cells, making the heart a prime target for the virus [39]. Interestingly, the binding of SARS-CoV-2 with ACE2 receptor causes internalization and loss of ACE2, resulting in unopposed angiotensin II activity leading to cardiac hypertrophy, fibrosis, reactive oxygen species production, and ultimately myocardial cell apoptosis [40,41,42]. In addition, studies found that cardiomyocytes express SARS-CoV-2 and coronavirus-associated receptors and factors (SCARFs), cathepsin-L, and furin that may facilitate direct cellular invasion by SARS-CoV-2 [43,44]. Pluripotent stem cell-derived cardiomyocytes infected with SARS-CoV-2 have been shown to have increased interferon signaling, apoptosis, and reactive oxygen stress which result in reduced cardiac contractility and cell death [45]. In vitro studies have demonstrated that SARS-CoV-2 can also infect the stromal cells and cause further cardiotoxicity owing to the differentiation of stromal cells into myofibroblasts. This will promote ACE2-dependent intracellular viral replication and, ultimately, the infection of adjacent cardiac cells [44]. It is further hypothesized that the formation of multinucleated syncytia may result in the fusion of contractile and non-contractile myocardial cells, ultimately causing further, extensive cardiac injury [46]. Finally, a viral infection of the endothelial cells and pericytes leads to increased platelet activation and microthrombi formation within the coronary circulation. These provoke ischemic myocardial injury, fibrosis, and ultimately, cardiac dysfunction [47].

Infection of the type-2 pneumocytes and the respiratory epithelium results in robust pro-inflammatory response and IL-6 secretion [48]. This causes increased differentiation of T-helper 17 cells (TH17), increased production of cytotoxic T-cells, and increased B-cell differentiation [49]. Previous autopsy studies of patients with fatal COVID-19 infection have demonstrated myocardial infiltration by mononuclear cells suggesting the possibility of immune-mediated myocardial damage in the setting of polyclonal T-cell activation [50]. In cases of severe SARS-CoV-2 infection, the host may develop an immune-mediated systemic hyperinflammatory response (cytokine storm) that can potentially provoke myocarditis in affected patients [51]. Interestingly, chronic exposure to IL-6 was shown to aggravate viral myocarditis underpinning the potential role of cytokines and immune system dysregulation in the pathophysiology of COVID-19-associated myocarditis (Figure 1) [52].

## 4. Serum Biomarkers in the Diagnosis of COVID-19-Associated Myocarditis

### 4.1. Cardiac Troponins

Troponins are regulatory proteins localized in the cytoplasm of cardiac myocytes that facilitate muscle contraction. Of the three subunits of the troponin complex, diagnostic laboratory tests measure the serum levels of troponin I (TnI) and troponin T (TnT), primarily to establish the diagnosis of acute myocardial infarction [53]. In addition, peak values were shown to correlate strongly with infarct size and clinical outcomes [54,55,56]. However, with the increasing sensitivity of the assays utilized, a broad range of other potential etiologies for elevated serum troponin values were described. These may include myocarditis, infiltrative cardiomyopathies (such as cardiac sarcoidosis and amyloidosis), cardiac contusion, pericardial diseases, pulmonary embolism, in addition to the infectious, immune, renal, nervous system, musculoskeletal, and inherited causes [57,58,59,60,61,62,63,64,65,66,67,68,69]. Given their high sensitivity and specificity, serum troponin levels were assayed routinely in patients with SARS-CoV-2 infection to screen for biochemical evidence of cardiac involvement. Studies have suggested 14%–36% incidence for direct or indirect myocardial injury in patients hospitalized due to COVID-19, depending on disease severity [29,70,71]. Typical peak troponin levels are significantly below those expected for acute myocardial infarction, despite evidence of inflammation detected on cardiac magnetic resonance imaging (MRI) [72,73]. In patients with COVID-19-associated myocarditis, the typical range for high sensitivity TnT was 0.103–0.157 ng/mL (normal: <0.014 ng/mL), while for high sensitivity TnI, it ranged from 1.34 to 2.54 ng/mL (normal: <0.04 ng/mL) [74,75]. Similar to ischemic myocardial injury, higher values are a strong predictor of ICU admission, in-hospital as well as 30-day mortality [76,77,78,79,80,81]. These observations emphasize the utility and importance of serum troponin measurements to diagnose myocarditis, identify individuals who may require early treatment escalation, and for prognostication purposes [82,83]. In addition, among patients recovered from COVID-19 infection, a history of elevated serum troponin was associated with a high incidence of myocarditis-like scarring on cardiac MRI, but without notable edema. This suggests that the myocardial injury caused by SARS-CoV-2 or the provoked immune response may be permanent, although we are lacking long-term follow-up data at this time [84]. Scar development in the ventricles may lead to future cardiovascular complications such as heart failure with reduced ejection fraction, or fatal arrhythmias [85]. However, a longer observational period is needed to validate these concerns and there are several studies currently underway.

### 4.2. Circulating Natriuretic Peptides

The natriuretic peptide family encompasses several structurally similar peptides, including the B-type (BNP) and N-terminal pro-B-type natriuretic peptide (NT-proBNP). The 108 amino acid pre-hormone, ProBNP, is synthesized in the ventricular myocardium and is released into the circulation primarily in response to hemodynamic stress [86,87]. Other factors that may stimulate its secretion include myocardial ischemia, circulating cytokines, and neurohormonal activation. Subsequently, ProBNP is cleaved into the biologically active BNP hormone and the inactive fragment, NT-proBNP. Their serum level can be assayed routinely and, since their first description in 1988, they became central to the diagnosis of patients with heart failure [87] In addition, they guide clinical patient management decisions, predict mortality, and have high prognostic accuracy [87,88].

Given their cardiac specificity, the utility of measuring circulating natriuretic peptide levels in patients with COVID-19 disease was evaluated by several groups. Elevated serum NT-proBNP concentration in this population has independently been associated with prolonged hospitalization, need for mechanical ventilation, increased rate of mortality, and worse overall prognosis, independent of prior history of heart failure [89,90,91,92,93,94]. Accurately identifying patients at the highest risk for poor outcomes can assist with triage decisions and facilitates resource allocation during times of critical demand. While the exact pathophysiological pathways remain to be elucidated, multiple mechanisms are postulated to contribute to the elevated NT-proBNP level in the setting of SARS-CoV-2 infection. These may include the systemic inflammatory response, hypoxia due to severe pneumonia, volume overload, bacterial superinfection/sepsis, and microvascular thrombi formation [89,90]. Although through different pathways, these conditions provoke increased myocardial strain prompting the synthesis and release of ProBNP from the ventricular cardiomyocytes. Given its recognized clinical utility, studies have recommended obtaining cardiac imaging among patients with COVID-19 disease and elevated age-specific circulating NT-proBNP levels to assess the degree of cardiac involvement [90,95]. Identifying stress cardiomyopathy or myocarditis early in the disease process may alter clinical decision-making and patient management strategies.

### 4.3. Other Potential Biomarkers to Detect Myocarditis in COVID-19 Infection

Given their relative ease to obtain and the lack of additional risk to hospital staff, there is a constant search for disease-specific biomarkers to improve diagnostic sensitivity, specificity, to identify patients at the highest risk, and to predict outcomes. In addition to cardiac troponins and natriuretic peptides, several studies have demonstrated a close association between elevated C-reactive protein (CRP), d-dimer, IL-6, and lactate dehydrogenase (LDH) levels and the clinical severity of SARS-CoV-2 infection as well as mortality [71,78,96,97]. Changes in the same biomarker concentrations have specifically been evaluated in individuals with COVID-19-associated myocarditis. Not surprisingly, all biomarkers were significantly elevated in the majority of patients. In addition to their diagnostic utility, these also provided incremental information for prognostication [19,27,98]. It is important to emphasize, however, that many of these changes may be related to the underlying systemic inflammatory process in this population. Despite their high sensitivity, the specificity of these biomarkers for myocarditis is limited [74].

## 5. Histopathological Testing for COVID-19-Associated Myocarditis

Histopathological analyses are fundamental to understanding the pathophysiological mechanisms of myocardial injury and myocarditis in COVID-19. Our initial experience stems from postmortem autopsy studies and although results were very diverse, many described the presence of inflammatory infiltrates in the myocardium and SARS-CoV-2 RNA or viral particles were also identified by various methodologies [76,99,100,101,102]. However, the presence of viral particles was not always associated with significant inflammation [103]. In contrast, other investigators failed to detect evidence for the viral presence in the cardiac tissue but identified an abundance of inflammatory cells [104,105].

Halushka and Vander Heide published a literature review on 277 postmortem COVID-19 autopsies and found that myocarditis was overall rare. It was present in 7.2% of examined hearts, however, most cases were felt to be “functionally insignificant”, and the actual prevalence of myocarditis was below 2% per the authors [106]. According to this review, the most common findings in cardiac tissue in COVID-19 patients were single-cell ischemia (13.7%) and non-myocarditis inflammatory infiltrates (12.6%) followed by myocarditis (7.2%) and acute myocardial infarctions (4.7%) [106]. Another study of 21 autopsies showed evidence of lymphocytic myocarditis in three (14%) of the cases–two of these were CD4 predominant and in one case CD8 predominant. In 86% of these cases, increased interstitial macrophage infiltration was present [101].

The highly diverse histopathologic findings on autopsy studies suggest potential observation biases, inconsistent reporting, but may also be related to substantial differences in SARS-CoV-2 viral load, underlying comorbid conditions, immune response, and time between the initial infection and death. Another potential cause of “over diagnosis” of COVID-19 associated myocarditis is the presence of tissue lymphocytes in older hearts, which may be attributed to myocarditis despite the absence of tissue necrosis. This was specifically noted to be the case in a study from Finland of autopsies in non-COVID-19 myocarditis cases [107].

When evaluating for cardiac involvement in the setting of COVID-19 infection in clinical practice, endomyocardial biopsy (EMBx) may provide a definitive diagnosis and remains the gold-standard diagnostic tool [108,109]. It may be performed safely in experienced centers with an overall very low complication risk. However, transferring patients to these centers has been challenging in the US and worldwide due to capacity constraints and the paucity of available beds, especially in the intensive care units. The diagnosis of myocarditis is based on the Dallas Criteria that require the presence of inflammatory infiltrates (including lymphocytes and macrophages) associated with myocyte necrosis of non-ischemic cause and interstitial edema [27,106]. However, it is important to note these findings are not exclusive to SARS-CoV-2 infection and may be present in myocarditis cases caused by other etiologies. Overall, the additional clinical yield of routine EMBx is limited as diagnosis may be made non-invasively with high certainty based on clinical presentation, serum biomarker evaluation, and the results from multimodality imaging [27,108].

## 6. Imaging Studies to Detect and Monitor COVID-19-Associated Myocarditis

Transthoracic echocardiography (TTE) remains the first-line imaging modality for the assessment of COVID-19-associated myocarditis. It is widely available, relatively inexpensive, and provides rapid, non-invasive assessment of cardiac structure and function. It not only allows for early diagnosis of cardiac involvement but also enables longitudinal monitoring throughout the disease course and recovery phase [110]. Echocardiographic findings characteristic to fulminant myocarditis include severely reduced biventricular function with preserved diastolic chamber dimensions, the increased ventricular wall thickness in the setting of myocardial edema, pericardial effusion, and potential intracavitary thrombus formation [111,112,113,114]. It is important to emphasize, however, that several investigators described similar TTE findings, including severe ventricular dysfunction and pericarditis, in patients infected with SARS-CoV-2 but without biochemical evidence of myocarditis [115,116,117]. This is most likely related to the severe systemic inflammatory response in the setting of COVID-19 in these patients. Biventricular mechanics as characterized by strain imaging is able to identify subtle changes in left ventricular function, even when it is below the detection limit for conventional two-dimensional TTE [118]. However, the finding of abnormal myocardial strain is non-specific, it is not indicative of the underlying etiology. When present in cases of myocarditis, it correlates well with the severity of myocardial edema [119]. The clinical utility of global longitudinal strain imaging was reported previously among patients with COVID-19 disease [120,121]. Most likely owing to the severe pneumonia caused by SARS-CoV-2 and the consequently increased afterload, right ventricular global, and free wall strain were reported to be significantly decreased in some patients and it showed a good correlation with poor clinical outcomes [120,121]. In contrast, left ventricular global longitudinal strain was often reduced in this population, regardless of outcomes [121]. The overlap in echocardiographic findings highlights the importance of multimodality imaging supplemented by serum biomarker testing when evaluating this patient population, especially when myocarditis is suspected [122].

Cardiac MRI is the non-invasive gold standard imaging modality for myocardial tissue characterization owing to its ability to reliably detect edema and fibrosis [123]. It also allows for detailed assessment of the pericardium, which is important as pericardial involvement often affects prognosis and management [124,125]. Given the poor outcomes in the presence of cardiovascular involvement with COVID-19 infection, cardiac MRI has been utilized frequently in this population to objectively assess the prevalence of myocarditis and to determine the extent of myocardial injury [126,127]. This provides invaluable information complementing clinical and biomarker data. The diagnosis of myocarditis by MRI may be established based on the 2018 Lake Louise Criteria (LLC) with 88% sensitivity and 96% specificity [128,129]. It requires at least one edema-sensitive criteria (T2-weighted images or T2 mapping) combined with at least one T1-based tissue characterization technique (late gadolinium enhancement, T1 mapping, or extracellular volume). Supportive findings include the presence of pericardial effusion and left ventricular systolic dysfunction [128]. In patients with COVID-19-associated cardiac injury, the most common pathway to fulfill LLC was through T1 and T2 mapping [109,126,127]. Compared to classic myocarditis, fewer patients had late gadolinium enhancement or decreased ventricular function [130]. The LLC criteria were used in a large-scale cohort study enrolling competitive college athletes with COVID-19 infection to screen for myocarditis [131]. Clinical or subclinical myocarditis was diagnosed in 2.3%, while only 0.31% would have been diagnosed based on clinical symptoms alone, underpinning the diagnostic sensitivity of MRI. Overall, compared to classic myocarditis, cardiac MRI findings were less pronounced in cases of COVID-19-associated myocarditis [109]. Despite its high accuracy in myocardial assessment, cardiac MRI has multiple disadvantages that limit its widespread and routine use. These include limited overall availability, high incidence of patient claustrophobia, prolonged exam duration, contraindication to scanning with certain implanted metallic devices, and inability to perform in critically ill patients [123].

Cardiac multidetector computed tomography (MDCT) is useful in the work-up for myocarditis. It has the ability to rule out significant epicardial coronary artery disease as the cause of clinical symptoms and elevated cardiac-specific biomarkers [132]. MDCT can also measure myocardial extracellular volume, an established marker for edema, and provides results that are in close agreement with data obtained with cardiac MRI [133]. Similarly, MDCT was found to be comparable to MRI in assessing myocardial delayed enhancement with the use of the monochromatic 70 keV technique [134]. Although not routinely utilized as of now, MDCT may have a role in myocardial assessment when cardiac MRI cannot be obtained. Its broad availability in medical centers and the shorter image acquisition times also confer an advantage to MDCT [135].

Nuclear imaging with radiolabeled Tc-99 m MIBI is used routinely to assess myocardial perfusion, myocardial cell viability, and membrane integrity by determining tracer uptake and clearance [118]. Previous studies have also documented abnormally reduced tracer uptake in the setting of myocardial inflammation and necrosis [136,137]. However, nuclear imaging is not used routinely in the diagnostic workup for myocarditis due to multiple limitations. These include high radiation exposure, prolonged acquisition time, high testing cost, and overall limited availability [138].

18F-fluorodeoxyglucose (18F-FDG) positron emission tomography combined with CT (PET-CT) can reliably quantify the degree of inflammation and may be considered as an alternative modality for patients with contraindications to cardiac MRI [118]. It has a sensitivity of 74% and specificity of 97% in diagnosing myocarditis [139,140]. Despite these potential benefits, it is not used commonly in the assessment of myocarditis due to the extensive patient dietary preparations required by the study protocol. Studies are currently underway to further describe the utility of PET-CT in acute myocarditis [118].

## 7. Myocarditis in COVID-19 Vaccine Recipients

Emergency use authorizations in the US were first granted to two COVID-19 vaccines in December 2020 and January 2021. Since their widespread introduction to the general public, many adverse events were reported. Data collected in the US focus primarily on COVID-19 vaccines developed by Pfizer-BioNTech (Fosun Pharma, Shanghai, China), Moderna (Cambridge, MA, USA), Oxford-AstraZeneca (Oxford, United Kingdom), Johnson & Johnson (New Brunswick, NJ, USA), and Sinovac Biotech (Beijing, China). Several cardiovascular complications are listed among the adverse events of special interest published by the Food and Drug Administration Center for Biologics Evaluation and Research, including myocarditis, myopericarditis, myocardial infarction, and stress cardiomyopathy with myocarditis being the most common [141]. Myocarditis and pericarditis had the highest reported incidence among Pfizer-BioNTech and Moderna vaccine recipients, however, these were overall very rare events at less than 10 per 100,000 people [141,142]. Given the high morbidity and mortality associated with COVID-19 infection, the benefit of vaccination clearly outweighs the minimal risk of severe complications. However, providers should remain vigilant to recognize early signs of post-vaccine myocarditis and research should continue to better elucidate the potential underlying mechanisms, some of which are reviewed briefly below.

One proposed mechanism for vaccine-related myocarditis is the activation of the body’s innate immune response by exogenous particles found in the messenger RNA (mRNA) vaccines. Although modifications are made to the nucleoside material, it remains immunogenic by activating Toll-like receptors (TLRs) and dendritic cells, thereby initiating pro-inflammatory cascades [142,143]. In a limited number of individuals with genetic predisposition, these may provoke tissue damage at the myocardial level [142,143]. In addition, the lipophilic material used to facilitate the cellular uptake of vaccine content was described in case reports to precipitate an inflammatory response [141].

Another potential mechanism for both COVID-19 and vaccine-associated myocarditis is the molecular mimicry between the spike protein of SARS-CoV-2 and self-cardiac antigens, such as myosin. This may trigger pre-existing, dysregulated immune pathways in predisposed individuals leading to polyclonal B-cell expansion, immune complex formation, and inflammatory response targeting cardiac proteins [143,144].

A third potential mechanism also involves the S glycoprotein of SARS-CoV-2. Structural studies have revealed that it binds to ACE2 with high affinity [145,146,147]. Among other roles in the human body, ACE2 degrades angiotensin II, a protein associated with inflammation. The S protein introduced into the cells by the mRNA vaccines bind ACE2, thereby preventing the degradation of angiotensin II. This may contribute to the inflammatory process, specifically in the myocardium [141]. To the best of our knowledge, this theory has not yet been examined in de novo studies.

## 8. Conclusions

This review aims to summarize the evolving scientific knowledge pertinent to the incidence, pathophysiology, and diagnostic evaluation of COVID-19-associated and post-vaccination myocarditis. The most recent data suggest that, despite a substantially increased risk compared to individuals without COVID-19 infections, the overall incidence of myocarditis in the setting of SARS-CoV-2 infection is lower than described by case reports and case series early in the pandemic. We reviewed the diagnostic modalities, both biomarkers and imaging, that may be used to diagnose and monitor myocarditis in this population. These are summarized in Table 1. It will be important to establish diagnostic criteria specific for COVID-19-associated myocarditis in order to differentiate this entity from other cardiac complications of the viral infection, such as stress cardiomyopathy, myocardial ischemia due to underlying coronary disease or thrombotic events, or sepsis-related myocardial dysfunction. For example, several commentaries question whether simply meeting LLC on cardiac MRI is indeed diagnostic for COVID-19-associated myocarditis [109,131]. Further research is warranted in this area, with studies potentially correlating non-invasive findings with results noted on EMBx. In addition, it is imperative that we develop strategies to monitor survivors for long-term disease sequelae.

It is critical for the scientific community to stay abreast of the most recent data on the cardiovascular complications of COVID-19 given the high associated morbidity and mortality. Our understanding of myocarditis in the setting of SARS-CoV-2 infection has evolved significantly since the beginning of the pandemic and is expected to progress further in the future. Maintaining scientific curiosity and continuing research related to this new field remains critically important so that we can engineer more effective vaccines, implement successful preventive strategies, and develop novel medical therapies.

## Figures and Tables

**Figure 1 biology-11-00520-f001:**
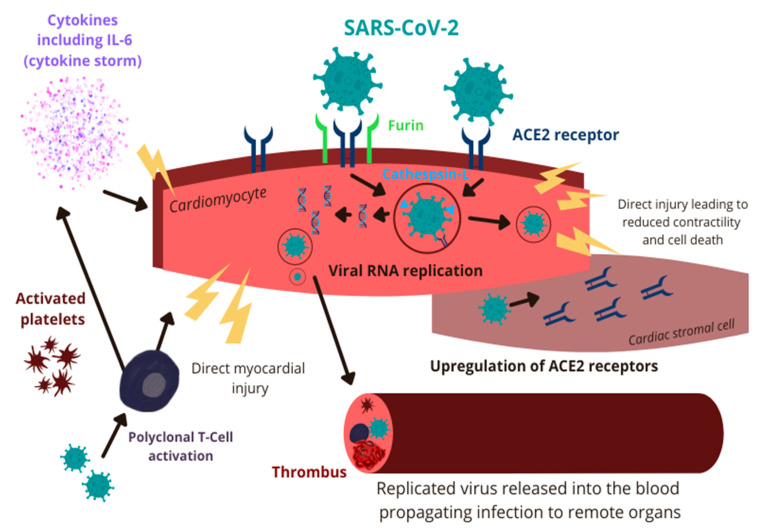
Pathophysiology of COVID-19-associated myocarditis. SARS-CoV-2: severe acute respiratory syndrome coronavirus 2; IL-6: interleukin-6; ACE2: angiotensin-converting enzyme 2; RNA: ribonucleic acid.

**Table 1 biology-11-00520-t001:** Diagnostic modality, expected findings, and relative specificity for COVID-19-associated myocarditis. “+” represents non-specific finding while “+++++” corresponds to high specificity.

Diagnostic Modality	Expected Finding in COVID-19 Associated Myocarditis	Relative Specificity
Cardiac troponin	Elevated [70,71]	+++
Brain-type natriuretic peptide	Elevated [78,80,81]	++
C-reactive protein	Elevated [78,80,81]	+
Interleukin-6	Elevated [78,80,81]	+
Lactate dehydrogenase	Elevated [78,80,81]	+
Transthoracic echocardiogram	LV dysfunction, normal LVIDd, increased wall thickness, pericardial effusion, possible LV thrombus [111,112,113,114,115,117,118,123]	++
Cardiac MRI	Presence of LGE, edema, LV dysfunction, possible pericardial effusion [126,127]	++++
Cardiac multidetector CT	Increased myocardial extracellular volume [133,134]	+++
Endomyocardial biopsy	Interstitial edema, lymphocytic infiltrate, increased macrophage presence, myocyte necrosis [76,100,101,102,103,114]	+++++

CT: computer tomography; LV: left ventricular; LVIDd: left ventricular internal diameter in diastole; LGE: late gadolinium enhancement; MRI: magnetic resonance imaging.

## Data Availability

Not applicable.

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
