# Peer review of "COVID-19-Associated Myocarditis: An Evolving Concern in Cardiology and Beyond"

_biology, 2022, doi:10.3390/biology11040520_

Round 1

Reviewer 1 Report

Daer Authors thank you on behalf of scientific community for this excellent work.

Despite it is a only narrative review in a context of thousands of covid 19 narrative papers this summarizes some very important clinical points. The main author (being english speaking) has well constructed grammatically the work.

LINE 179-182: this is true but it is the same in all infection diseases morover please explain the concept mortality-bnp in the edge of coronavirus.

For the rest paper you are in the same line of the rest of scientific world.

Please consider this works to add at your work:

Zanza C, Romenskaya T, Manetti AC, Franceschi F, La Russa R, Bertozzi G, Maiese A, Savioli G, Volonnino G, Longhitano Y. Cytokine Storm in COVID-19: Immunopathogenesis and Therapy. Medicina (Kaunas). 2022 Jan 18;58(2):144. doi: 10.3390/medicina58020144. PMID: 35208467; PMCID: PMC8876409.

Piccioni A, Saviano A, Cicchinelli S, Franza L, Rosa F, Zanza C, Santoro MC, Candelli M, Covino M, Nannini G, Amedei A, Franceschi F. Microbiota and Myopericarditis: The New Frontier in the Car-Diological Field to Prevent or Treat Inflammatory Cardiomyo-Pathies in COVID-19 Outbreak. Biomedicines. 2021 Sep 16;9(9):1234. doi: 10.3390/biomedicines9091234. PMID: 34572420; PMCID: PMC8468627.

Author Response

Thank you for your review! With your feedback, we added a section on natriuretic peptides. We have also added content from both references as recommended. 

Reviewer 2 Report

The review by Meg Fraser and colleagues gives a detailed outlook of the recent (or ongoing) pandemic - COVID-19 with respective to the origin of the virus, life cycle, spread, complication and efforts towards immunization. Specifically, the article focusses on the less understood COVID associated myocarditis. The authors extensively review the literature and provide an educated opinion on how it is caused and what are the features that can be used for early detection. These compiled observations provide the much-needed toolkit to successfully combat COVID-19 associated complications.

To address the broad readership of the journal and the significance of the current review, it will be beneficial to incorporate a schematic of the virus life cycle and its interactions with cardiomyocytes (as described in section 3). Also, it will be useful to add how the vaccine mimics these interactions (from section 7).

Author Response

Thank you very much for your feedback! We have added a figure on the pathophysiology of COVID-19 myocarditis as described in the text.

Reviewer 3 Report

In the review article “COVID-19-associated myocarditis: An evolving concern in cardiology and beyond” Fraser et. al., have summarized the prevalence of COVID-19-associated and post-vaccination myocarditis, as well as their pathophysiology, and diagnostic evaluation. Furthermore, authors endorsed the overall incidence of COVID-19 associated myocarditis is lower than described by case reports performed at the early stage of the pandemic. Additionally, a table describing the diagnostic modalities, expected findings, and relative specificity for COVID-19-associated myocarditis is included. This review is systematically organized with relevant information, which is going to enrich the existing knowledge to the arena of research related to COVID-19-associated myocarditis; hence, it may be accepted for publication after addressing following points. presented

  1. Subsection “3.2” (line-175) appeared after 4.1. This should be corrected.
  2. Heading for section “4” and “5” are same “Serum biomarkers in the diagnosis of COVID-19-associated myocarditis”. Please make sure that it is rectified in the final version of the article.
  3. “Our initial experience stems from postmortem autopsy studies” line-191 is incomplete.
  4. Since death toll due to COVID-19 is increasing day-by-day, therefore time point should be mentioned in the sentence “The estimated worldwide COVID-19-related death toll has surpassed an astonishing 5.6 million” line-45.

Plagiarism percentage - not checked by the reviewer.

Author Response

Thank you very much for your feedback. We appreciate the time to review our manuscript. We have addressed your comments and corrections as below:

  1. Subsection “3.2” (line-175) appeared after 4.1. This should be corrected. Thank you for your correction, this has been fixed. 
  2. Heading for section “4” and “5” are same “Serum biomarkers in the diagnosis of COVID-19-associated myocarditis”. Please make sure that it is rectified in the final version of the article. Thank you for your correction, this has been fixed.
  3. “Our initial experience stems from postmortem autopsy studies” line-191 is incomplete. Thank you, we have editing this sentence as follows: "Our initial experience stems from postmortem autopsy studies and although results were very diverse, many described the presence of inflammatory infiltrates in the myocardium and SARS-CoV-2 RNA or viral particles were also identified by various methodologies."
  4. Since death toll due to COVID-19 is increasing day-by-day, therefore time point should be mentioned in the sentence “The estimated worldwide COVID-19-related death toll has surpassed an astonishing 5.6 million” line-45. Thank you for this feedback. We have added "as of February 2022 and continues to increase".